# Discriminant validity of 3D joint kinematics and centre of mass displacement measured by inertial sensor technology during the unipodal stance task

R. van der Straaten[1]*, M. Wesseling[2], I. Jonkers[2], B. Vanwanseele[2], A. K. B. D. Bruijnes[3], J. Malcorps[4], J. Bellemans[3], J. Truijen[3], L. De Baets[1], A. Timmermans[1]

**1** REVAL Rehabilitation Research Center, Hasselt University, Diepenbeek, Belgium, **2** Department of Movement Sciences, Human Movement Biomechanics, KU Leuven, Leuven, Belgium, **3** Department of Orthopaedics, Ziekenhuis Oost-Limburg, Genk, Belgium, **4** Department of Orthopaedic Surgery, Jessa Hospital, Hasselt, Belgium

☉ These authors contributed equally to this work.
* rob.vanderstraaten@uhasselt.be

**Data Availability Statement:** Data files are available from the OSF database at (https://osf.io/3gb6m/).

## Abstract

### Background

The unipodal stance task is a clinical task that quantifies postural stability and alignment of the lower limb joints, while weight bearing on one leg. As persons with knee osteoarthritis (KOA) have poor postural and knee joint stability, objective assessment of this task might be useful.

### Objective

To investigate the discriminant validity of three-dimensional joint kinematics and centre of mass displacement (COM) between healthy controls and persons with knee KOA, during unipodal stance using inertial sensors. Additionally, the reliability, agreement and construct validity are assessed to determine the reproducibility and accuracy of the discriminating parameters.

### Methods

Twenty healthy controls and 19 persons with unilateral severe KOA were included. Five repetitions of the unipodal stance task were simultaneously recorded by an inertial sensor system and a camera-based system (gold standard). Statistical significant differences in kinematic waveforms between healthy controls and persons with severe knee KOA were determined using one-dimensional statistical parametric mapping (SPM1D).

### Results

Persons with severe knee KOA had more lateral trunk lean towards the contralateral leg, more hip flexion throughout the performance of the unipodal stance task, more pelvic obliquity and COM displacement towards the contralateral side. However, for the latter two

**Funding:** This research is part of the Limburg Clinical Research Center (LCRC) UHasselt-ZOL-Jessa, supported by the foundation Limburg Sterk Merk, province of Limburg, Flemish government, Hasselt University, Jessa Hospital and Ziekenhuis Oost-Limburg. The organizations did not have a contribution to the creation of this manuscript.

**Competing interests:** The authors have declared that no competing interests exist.

parameters the minimum detectable change was greater than the difference between healthy controls and persons with severe knee KOA. The construct validity was good (coefficient of multiple correlation 0.75, 0.83 respectively) and the root mean squared error (RMSE) was low (RMSE <1.5˚) for the discriminant parameters.

## Conclusion

Inertial sensor based movement analysis can discriminate between healthy controls and persons with severe knee KOA for lateral trunk lean and hip flexion, but unfortunately not for the knee angles. Further research is required to improve the reproducibility and accuracy of the inertial sensor measurements before they can be used to assess differences in tasks with a small range of motion.

## Introduction

Knee osteoarthritis (KOA) is a common degenerative joint disease in middle-aged and older adults, with up to 18% of persons above 60 years showing symptoms of KOA [1]. Reduced range of motion (ROM), proprioception, muscle strength, and increased joint load are frequently associated with degenerative changes in the joint, together with the development of pain and functional limitations [1, 2]. Another important factor, present in 44 to 72% of persons with KOA, is self-reported knee joint instability, which is defined as the sensation of buckling, shifting or giving away of the knee [3–5]. After a total knee replacement (TKR), one third of the persons complains of knee joint instability at six months after surgery [4]. Since self-reported knee instability is associated with a reduced physical functioning and a decreased performance of daily life activities, it is essential to evaluate knee joint instability, next to pain, muscle strength and joint ROM in persons with KOA during the course of their treatment [3–6].

Objective measures to quantify knee joint (in)stability, ready to be used in a clinical setting, are currently not described. Previous research investigated whether knee joint instability could be assessed by measuring knee varus/valgus movement during walking [7]. No relationship was established, as persons with KOA showed adaptive movement mechanisms at the trunk, hip and ankle to off-load the knee joint during walking. Since camera-based analysis of walking also requires dedicated lab space, its clinical application is limited. Therefore, other tasks that challenge the knee joint should be considered with regard to the assessment of knee instability [8]. The unipodal stance task requires that an individual maintains the knee in a given position, while sustaining a balanced body position over the fully loaded knee. Few studies investigated differences in knee joint stability of persons with and without KOA during an unipodal stance [9, 10]. These studies reported that the knee varus/valgus [9, 10] and the anterior-posterior (AP) and medio-lateral (ML) knee accelerations [10] were significantly higher in persons with KOA compared to age matched healthy controls. In addition, a significant correlation between the AP knee acceleration and centre of pressure sway was reported. This indicates that knee joint instability is indirectly linked to increased body sway [10].

Joint kinematics are generally measured in a movement laboratory, where movement can be assessed with high precision. Despite their high precision, these systems are expensive, complex to use and the measurements are time-consuming. Over the last decades, inertial sensor technology has gained popularity to be used for motion analysis, as they are less expensive, easy to use and less time consuming [11]. Besides joint kinematics, an inertial sensor system

provides information regarding the centre of mass (COM) displacement, which is defined based on the segment position and orientation, together with a body mass distribution model [12]. The COM is recognized as an estimate for postural sway [13]. That is why inertial sensors seem an ideal alternative for lab-based motion analysis in the objective assessment of the knee and adjacent lower limb joint kinematics in persons with KOA in clinical settings. However, before a system can be used in clinical settings, it is important to assess its validity and reliability. Therefore, the primary aim of this study is to investigate the discriminant validity of 3D joint kinematics and the COM displacement as measured by an inertial sensor system, by assessing differences in trunk and lower limb joint angles and postural sway (i.e. COM displacement) between healthy controls and persons with unilateral KOA during the unipodal stance. A secondary aim of this paper is to assess the within-session, between-session and between-operator reliability and agreement of the discriminant joint kinematics and COM displacement, together with their construct validity by comparing the outcomes of the inertial sensor system against these of a camera-based motion capture system (gold standard).

## Methods

### Participants

Twenty healthy participants were recruited from a local network of seniors and relatives. These healthy participants were included when they were between 50–75 years old, were able to walk 10 m, were able to ascent and descent a staircase of four steps, and when they understood the Dutch language. Participants were excluded if they experienced pain or pathology in the torso or lower limb joints, or had any systematic or neurological disease.

Additionally, 19 persons with unilateral KOA were recruited from two local hospitals: Jessa Hospital (Hasselt, Belgium) and Ziekenhuis Oost Limburg (Genk, Belgium). Persons with severe KOA were eligible for inclusion when the following criteria were met: age between 50–75 years old, diagnosis of unilateral KOA and awaiting for a total knee replacement (TKR) surgery, ability to walk 10 m and to ascent and descent a staircase of four steps, and ability to understand the Dutch language. Exclusion criteria were a corticosteroid injection in the knee during a period of three months before inclusion, diagnosis of degenerative disorders in other lower limb joints, neurological conditions or a history of pathological osteoporotic fractures. The outcome is a percentage in which 0% indicates extreme problems and 100% indicate no problems [14]. All participants gave informed consent before participation, as approved by the ethical committee of the academic hospital Leuven (reference no. s-59857).

### Study design

To evaluate reliability and agreement of the joint kinematics ROM and COM displacement by means of internal sensors, only healthy controls were included. This was a conscious decision, as patients' movement variability can affect day-to-day execution of the movement. To evaluate the construct and discriminant validity, only twelve healthy controls participated, because for these participants both legs were included in the analysis. These healthy legs were compared to the affected leg of the persons with KOA. Since this study is part of a larger project, justification of the number of subjects is based on an overview of compartmental forces measured in participants after TKR [15]. The study of Fregly and colleagues reported an average medial compartmental force of 1.61 (±0.305) body weight during gait. Assuming an increase of 1 Stdev (0.31 BW) to be clinically significant in subjects suffering from medial compartmental OA, a sample size of 14 subjects was calculated with a of 0.05 and power level of 0.80. As it was expected that some of the participants could dropout after inclusion some additional participants were recruited.

## Data collection

To evaluate the reliability and agreement, healthy controls visited the lab on two different days (5–20 days apart). On day one, the protocol was executed twice, to determine the within-session and between-operator reliability and agreement. On day two, participants returned to the lab and the entire procedure of day one was repeated, to evaluate the between-session reliability and agreement. As the reliability and agreement are part of a larger study, more details regarding the procedures are described elsewhere [16, 17]. The 12 healthy controls that participated in the validity study additionally performed the protocol on day two twice. Instead of removing the inertial sensors after the first session, optoelectronic markers were additionally positioned and 5 repetitions of the unipodal stance task were recorded simultaneously by the inertial sensor system (MVN BIOMECH Awinda, Vicon Technologies, Enschede, The Netherlands) and the optoelectronic system (VICON, Oxford Metrics, Oxford, UK).. The persons with severe KOA visited the lab only once.

The performance of the unipodal stance task was beforehand discussed with three orthopaedic surgeons (JM, JB and JT), and all of them agreed that it would be feasible for persons with severe KOA to perform the unipodal stance task. Moreover, all participants were beforehand carefully instructed regarding the performance of the unipodal stance task (Fig 1). Before task execution, the task was explained and executed by the operator that was guiding the measurements, to give the participant an indication of how to perform the task. Subsequently the task was practiced by the participant to familiarize a uniform task execution (according to the instruction given). The instructions were given by operators that were experienced with motion analysis studies, one of them was a physiotherapist with 12 years' experience and the other a human movement scientist with 7 years' experience. The unipodal stance task was performed barefoot and was practiced to familiarize a uniform task execution and to make sure it was executed according to the instructions. All participants were able to rest in between repetitions if required. Before each session, both groups completed the Knee injury and Osteoarthritis Outcome Score (KOOS) to evaluate the extent of symptomatic problems at the knee related to KOA.

The individual in this manuscript has given written informed consent (as outlined in PLOS consent form) to publish these case details.

| Task | | Instruction to participant |
|------|---|---------------------------|
| Unipodal stance |  | Stand still with the feet shoulder width apart and the hands on the pelvis. Shift the weight to one side of the body (i.e. stand on one leg) and lift the other foot from the ground. When standing on one leg, maintain balance for at least 3 seconds, before shifting the weight back over two legs and the foot is positioned on the floor. |

**Fig 1. Detailed description of the instructions to the participants.**

## Inertial sensor system

Trunk and lower limb joint kinematics and COM displacement were measured using 15 inertial sensors (MVN BIOMECH Awinda). The inertial sensors were positioned according to the guidelines of the manufacturer [18]. Trunk, pelvis, hip, knee and ankle angles were recorded using the MVN BIOMECH software (60 Hz, MVN Studio 4.4, firmware version 4.3.1). To scale the model, the participants' body dimensions were inserted and subsequently, a static (N-pose) calibration was performed to align the sensor to the segment. Three-dimensional joint kinematics and the COM displacement were directly derived from the MVN software, which were defined according to the recommendations of the international society of biomechanics [19]. The x-axis represents frontal plane joint movements (abduction/adduction), the y-axis the transversal plane joint movements (internal/external rotation), and the z-axis the sagittal plane joint movements (flexion/extension).

## Optoelectronic system

Three-dimensional marker trajectories were recorded using a 10 camera VICON System (100 Hz, Oxford Metrics, Oxford, UK). Therefore, 65 reflective markers were positioned according to the Plug-in-Gait model, with additional anatomical markers on the sacrum, medial femur epicondyles, medial malleoli and marker clusters on the upper and lower legs and arms [20]. Data was processed using a musculoskeletal model with 6 degrees of freedom (DOF) in the patellofemoral and tibiofemoral joints, 6 DOF for the pelvis, 3 DOF for the trunk hip joint and 1 DOF for the ankle joint [21]. This model was implemented in SIMM (Motion Analysis Corporation, Santa Rosa, CA), using the Dynamics Pipeline (Symbolic Dynamics, Inc, Mountain View, CA) and SD/Fast (PTC, Needham, MA) to generate the multibody equations of motion [22]. A generic model was scaled to the anthropometry and mass of the participant. Full body joint kinematics were calculated using inverse kinematics [23]. The COM was calculated as the summed position, weighted by the segmental mass of all the segments and was expressed in the antero-posterior (AP), medio-lateral (ML) and vertical direction.

## Data-analysis

For both systems, the joint kinematics and COM AP, ML and Vertical displacement were time normalized from 0–100% (from the period in which the foot was lifted more than 2 cm off the ground), using a custom written algorithm in Matlab (2016b, Mathworks, Inc., Natick, MA, USA). In order to compare the inertial sensor system with the camera-based system, trunk and pelvic angles were transformed to account for differences in the segment coordinate frames in the underlying kinematic models. This was required as within the musculoskeletal model the trunk was defined as one rigid body, whereas the trunk was divided into four segments in the MVN BIOMECH model. Therefore, these segments were accumulated to have a similar representation of the trunk in both models. Furthermore, the MVN BIOMECH pelvic orientation was converted into Euler angles to match the pelvic angles (expressed in the global reference frame) of the musculoskeletal model. Finally, the absolute displacement of the COM was compared, for both models. Therefore, COM displacement was normalized in all directions to its initial position at the start of the trial.

## Discriminant validity

To assess the discriminant validity, the entire waveform from both models were compared between the healthy controls and persons with severe KOA based on one-dimensional statistical parametric mapping (SPM1D) analysis. First, the normality of the waveforms was tested,

using the normality function of SPM for a two-sample t-test, which is based on the k2-residuals that evaluates skewness and kurtosis of the normal distribution [24]. After the normality was verified, dependent on the outcome a parametric two-sample t-test (SPM{t}, α = 0.05) or a non-parametric two-sample t-test (SnPM{t}, α = 0.05) was used. In case significant differences were present, SPM provided p-values for each time the t-curve exceeded the threshold of significance in the waveform.

### Construct validity

To evaluate the construct validity, waveforms from the camera-based system were compared with the waveforms of the inertial sensor system. Waveforms from healthy controls and persons with severe KOA were pooled, as the construct validity evaluates the difference between two waveforms, from both systems. Next, the normalized waveforms from both systems were compared using the root mean squared error (RMSE) and the coefficient of multiple correlation (CMC) [25]. As it is known that the CMC is not a real number (NaN) when the offset between both waveforms is comparable, the CMC was calculated after offset removal [25]. For further interpretation, the amount of NaNs and the mean and standard deviation (SD) are presented. The CMC was interpreted as follows: CMC >0.95 excellent, 0.85–0.94 very good, 0.75–0.84 good, 0.65–0.74 moderate and CMCs <0.64 low.

### Reliability and agreement

The reliability and agreement were determined from the inertial sensor data. Therefore, the range of motion (ROM) was calculated as the absolute difference between the minimum and maximum angle. The first trial was excluded from the analysis, as this trial could be disturbed by initiation strategies. All trials were visually examined for technical errors and only the right leg data was used for the analysis. The analysis of the reliability and agreement was performed using SPSS (version 25, IBM Corporation, Amonk, NY). Reliability was determined based on intraclass correlation coefficients (ICC), including the 95% confidence interval. Single data was used to calculate the within-session reliability ($ICC_{2,1}$) and agreement. Average data of four repetitions was used to calculate the between-session and between-operator reliability ($ICC_{2,k}$) and agreement. ICCs ≥ 0.90 were considered as excellent, 0.70–0.89 good, 0.69–0.40 acceptable, and <0.40 as low. Agreement was determined using the standard error of the measurement (SEM), based on the square root of the mean square error term of the analysis of variance (ANOVA) and the minimum detectable change (MDC) between two sessions, using the SEM (MDC = SEM × 1.96 × $\sqrt{2}$). To provide information on the magnitude of the SEM with respect to the ROM, a proportional SEM (%SEM) was calculated (%SEM = (SEM/mean)* 100%).

## Results

### Participants

Twelve healthy controls and 19 persons with severe KOA (unilateral KOA; Kellgren / Lawrence grade 3 (n = 1)– 4 (n = 18)) were recruited to investigate the discriminant & construct validity (Table 1). The KOA group was significantly older (p = 0.02), compared to the healthy group. No other significant differences were found in weight, height or BMI between groups. Additionally, the healthy controls did not show any symptomatic problems of the knee related to KOA according to the outcome of the KOOS questionnaire, as all subscales were close to 100 (i.e. indicates no problems) and the persons with severe KOA show significantly lower

**Table 1. Participant characteristics (mean ± SD).**

| | Reliability & Agreement | Discriminant & Construct validity | |
| --- | --- | --- | --- |
| | Healthy (n = 20) | Healthy (n = 12) | KOA (n = 19) |
| Male / Female | 9/11 | 6/6 | 12/7 |
| Age (years) | 62,7 (± 8,5) | 59,8 (± 7,0) | 65,1 (± 5,2) * |
| Height (m) | 1,70 (± 0,08) | 1,71 (± 0,10) | 1,75 (± 0,08) |
| Weight (kg) | 70,8 (± 14,9) | 74,3 (± 14,9) | 79,8 (± 8,4) |
| BMI | 24,3 (± 3,6) | 25,1 (± 3,4) | 26,0 (± 2,2) |
| KOOS Pain | 96,1 (± 5,9) | 95,1 (± 7,0) | 50,1 (± 12,2) * |
| KOOS Symptoms | 97,3 (± 5,2) | 98,5 (± 3,6) | 52,3 (± 18,4) * |
| KOOS ADL | 98,9 (± 2,4) | 98,7 (± 2,9) | 56,4 (± 15,9) * |
| KOOS Sport/Rec | 94,5 (± 8,7) | 94,6 (± 7,8) | 24,1 (± 23.9) * |
| KOOS QOL | 93.4 (± 9.4) | 94,8 (± 6,4) | 28.0 (± 16,3) * |

* Significant difference between healthy controls and persons with severe KOA (p<0.05).

scores on all subscales. The characteristics of the healthy controls who participated in the reliability and agreement study (n = 20) are added in Table 1.

## Discriminant validity

In both models it was shown that persons with severe KOA had significantly *more lateral trunk lean* towards the contralateral leg in the second half of the unipodal stance task (~50–100%; p = 0.018 and p-0.009) and *more hip flexion* throughout the unipodal stance task (0–100%; p = 0.001 and p = 0.001) in comparison to healthy control (Fig 2). Based on the camera-based system, persons with severe KOA had *more pelvic obliquity* at the contralateral side from 0 to 80% of the task (p = 0.010), whereas the inertial sensor system showed that persons with severe KOA had only more pelvic obliquity at the contralateral side from 0 to 21% (p = 0.009) of the unipodal stance task. Persons with severe KOA had significantly *more COM displacement* towards the contralateral side from 39 to 100% (p<0.001) according to the camera-based system, while this alteration was only found from 72 to 100% of the waveform (p = 0.004) with the inertial sensor system. Furthermore, differences between healthy controls and persons with severe KOA were observed from 0 to 100% of the *trunk flexion/extension* movement in both models. However, contradicting results were observed between systems, as based on the camera-based system persons with severe KOA had more trunk extension compared to healthy controls, and based on the inertial sensor system healthy controls had more trunk extension compared to persons with severe KOA (Fig 2).

Additionally, the camera-based system revealed that persons with severe KOA had more *knee adduction* between ~20–22% (p = 0.023) and from 46 to 100% (p = 0.001), more *trunk rotation* toward the standing leg from 92–100% (p = 0.023) and more *knee flexion* from 0 to 100% (p = 0.001) of the unipodal stance task. These differences were not identified by the inertial sensor system (Fig 3).

## Construct validity

The construct validity of the discriminating parameters ranged from moderate to good. The COM ML displacement had moderate construct validity (CMC 0.72, RMSE 0.008m). The trunk and pelvis ab/adduction waveforms (CMC 0.75, RMSE 1.2˚; CMC 0.81, RMSE 1.0˚ respectively) and hip flexion/extension waveforms (CMC 0.83, RMSE 0.7˚) had good construct validity. Although the CMCs were determined based on the mean corrected data, it remains

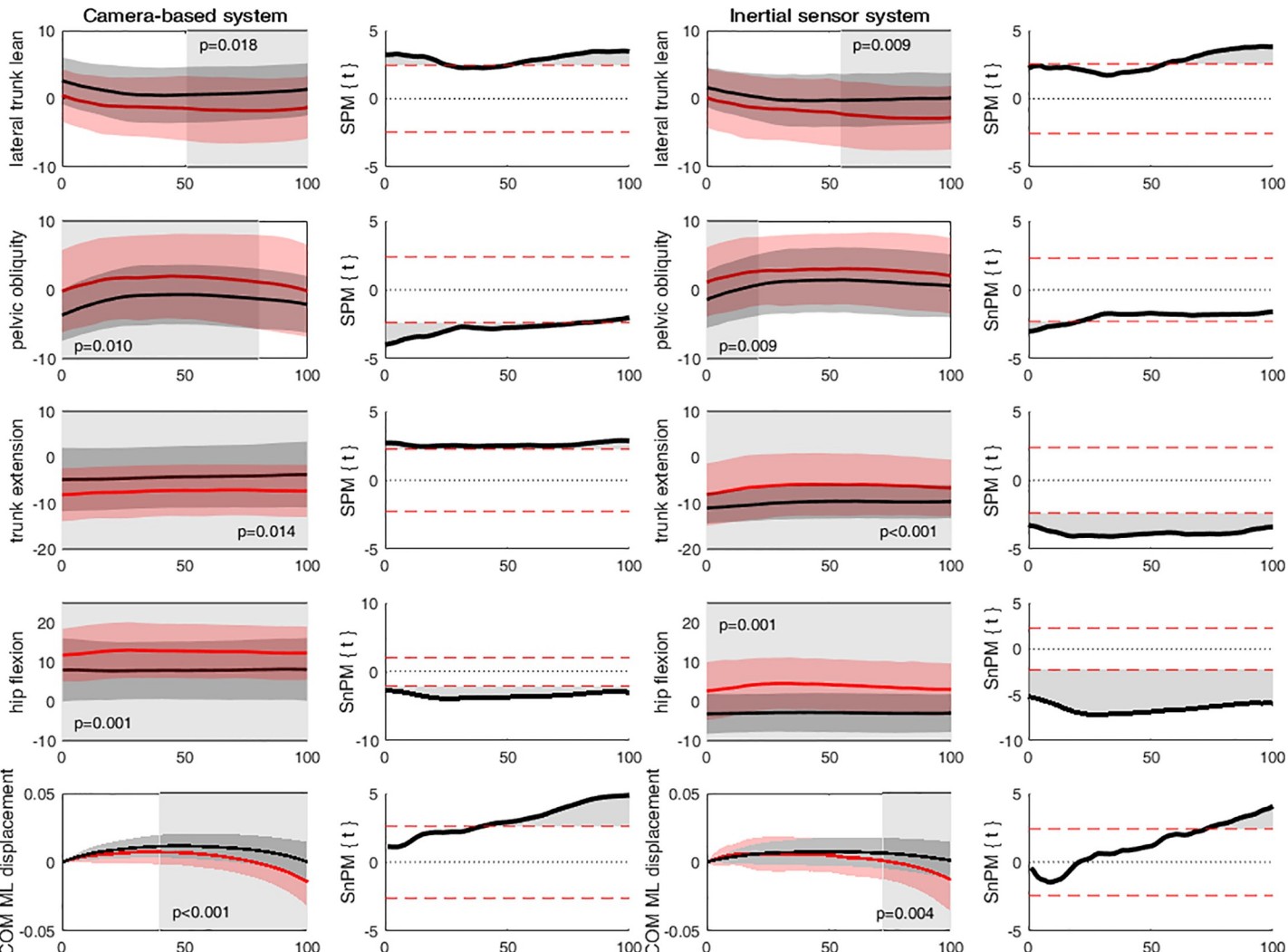

**Fig 2. Significant differences in joint kinematics and COM displacement between healthy controls (black) and persons with severe KOA (red) as identified by the camera-based system and the inertial sensor system using SPM1D.** SPM{t} was used for normally distributed waveforms and SnPM{t} for not normally distributed waveforms. The shaded area within the waveforms represents the area where the joint angles and COM displacement are significantly different (i.e. in which the SPM{t} / SnPM{t} exceeds the critical threshold).

impossible to calculate CMC for all the recorded trials. For the COM ML displacement, it was not possible to calculate a CMC for 24% of the data, for trunk ab/adduction this was not possible for 22% of the data, for pelvic ab/adduction for 15% of the data and for the hip flexion in 4% of the data. The CMCs and the corresponding RMSEs from all 3D joint kinematics and the corresponding waveforms from both systems are presented in the supplementary material (S1 Table, S1 Fig).

## Reliability and agreement

The within-session, between-session and between-operator reliability ranged from acceptable to good for the trunk and pelvis ab/adduction ROM (Table 2). Acceptable within-session reliability was observed for the hip flexion/extension ROM and the COM ML displacement. For these parameters, the between-session and between-operator reliability varied from low to

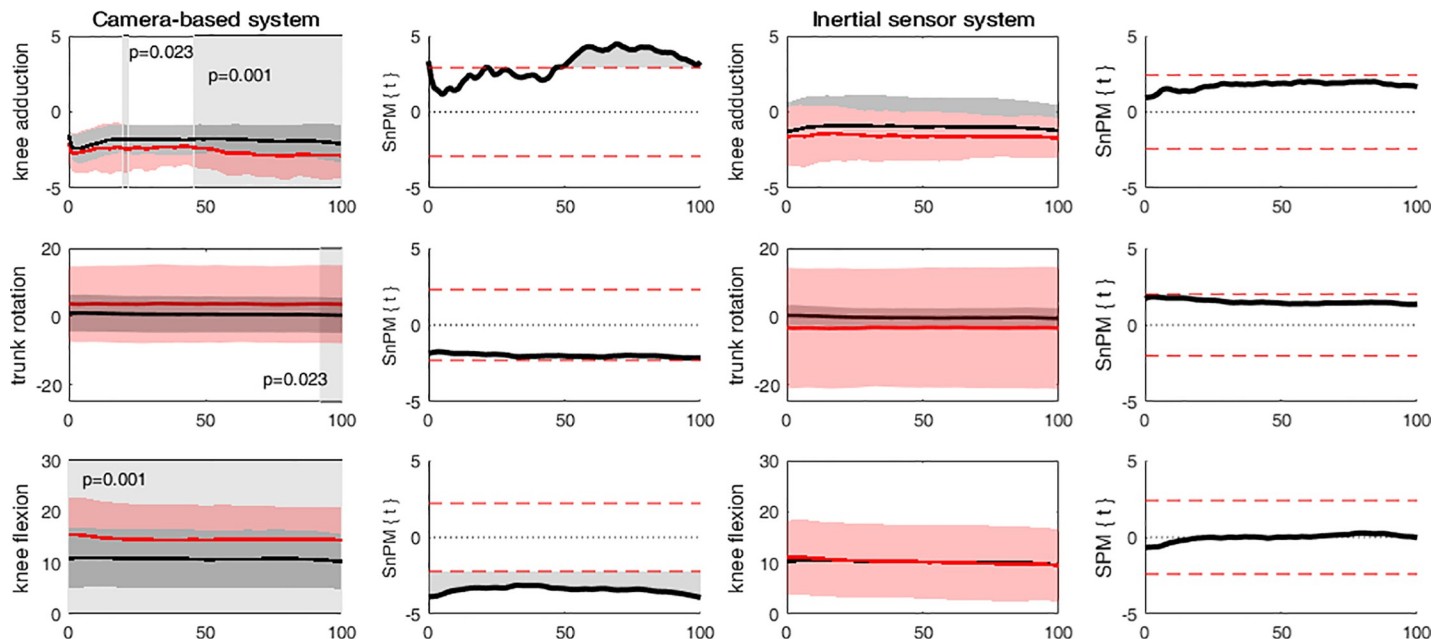

**Fig 3. Significant differences between healthy controls (black) and persons with severe KOA (red) as identified by the camera-based system only using SPM1D.** SPM{t} was used for normally distributed waveforms and SnPM{t} for not normally distributed waveforms. The shaded area within the waveform represent the area where the joint angles are significantly different (i.e. in which the SPM{t} / SnPM{t} exceeds the critical threshold).

good (Table 2). The agreement for the COM displacement ranged from 0.003m to 0.007m, therefore the proportional SEM ranged between 14% to 30%. For the trunk and pelvis ab/ adduction ROM and the hip flexion/extension ROM, the SEM ranged from 0.79 to 1.66˚. However, as the mean ROM in all joints was only ~4.0˚, the proportional SEM is relatively high (25–50%). Individual reliability and agreement results of the 3D joint ROMs and COM displacements are added in the Supplementary Materials (S2 Table).

**Table 2. Reliability and agreement of discriminating parameters in both models.**

| | | ICC | CI | mean (± SD) | SEM | MDC | %SEM | %MDC |
|---|---|---|---|---|---|---|---|---|
| Trunk—ab/ad | ws | 0,69 | 0,49–0,86 | 3,60 ± 1,90 | 1,04 | 2,88 | 28,9 | 80.0 |
| | bs | 0,71 | 0,25–0,89 | 3,46 ± 1,52 | 1,03 | 2,85 | 29,7 | 82.3 |
| | bo | 0,68 | 0,17–0,88 | 3,56 ± 1,37 | 0,93 | 2,58 | 26,2 | 72.6 |
| Pelvis—ab/ad | ws | 0,80 | 0,63–0,91 | 4,4 ± 2,4 | 1,02 | 2,83 | 23,4 | 64,8 |
| | bs | 0,76 | 0,37–0,91 | 4,0 ± 2,0 | 1,13 | 3,13 | 28,3 | 78,5 |
| | bo | 0,70 | 0,20–0,89 | 4,3 ± 1,9 | 1,28 | 3,54 | 29,7 | 82,2 |
| Hip—f/e | ws | 0,53 | 0,30–0,75 | 3,2 ± 1,4 | 0,97 | 2,68 | 30,1 | 83,3 |
| | bs | 0,29 | 0,00–0,72 | 3,5 ± 1,8 | 1,66 | 4,59 | 46,8 | 129,6 |
| | bo | 0,71 | 0,24–0,88 | 3,3 ± 1,2 | 0,79 | 2,20 | 24,3 | 67,5 |
| COM—ML | ws | 0,43 | 0,19–0,68 | 0,021 ± 0,010 | 0,007 | 0,018 | 30,5 | 84,6 |
| | bs | 0,81 | 0,51–0,92 | 0,027 ± 0,009 | 0,003 | 0,009 | 14,7 | 40,8 |
| | bo | 0,60 | 0,00–0,84 | 0,027 ± 0,008 | 0,004 | 0,012 | 21,6 | 60,0 |

CI: confidence interval, ws: within-session, bs: between-session, bo: between-operator, mean ± SD represent the average and SD ROM of the waveform, ab/ad: abduction/adduction ROM, f/e: flexion/extension ROM.

## Discussion

The aim of this study was to investigate the discriminant validity of 3D kinematics waveforms and COM displacement measured by an inertial sensor system during the unipodal stance task between healthy and persons with severe KOA. Initially, the discriminant validity was determined for the camera-based system, as this model is considered as the gold standard [26]. Eight discriminating parameters were identified in the frontal, transverse and sagittal plane and in the COM displacement. For four of these parameters, i.e. lateral trunk lean towards the contralateral leg, pelvic obliquity at the contralateral side, hip flexion and COM displacement towards the contralateral side, the inertial sensor system also discriminated between healthy and persons with severe KOA in the same direction. Additionally, the reliability, agreement and construct validity of these four discriminating parameters were investigated to verify whether these parameters could be measured in a reproducible and accurate manner. The reliability and agreement ranged from acceptable to good and the construct validity from moderate to good.

On average, persons with severe KOA had *a lateral trunk lean towards the contralateral leg* in the last 14% of the unipodal stance, while healthy controls maintained the trunk straight above the pelvis (i.e. trunk angle of zero degrees). Although a significant difference was observed in 50–100% of the waveform between healthy controls and persons with severe KOA this difference was only larger than the MDC from 86 to 100% of the waveform (Fig 2). It is assumed that persons with severe KOA have problems to maintain control at the end of the unipodal stance, while moving back to make contact with the ground again. Furthermore, it is suggested that persons with severe KOA perform a trunk lean towards the contralateral leg, instead of maintaining the trunk above the pelvis like the healthy controls, due to lack in trunk control. This was also reflected in the fact that persons with severe KOA show *greater COM displacement* towards the contralateral side from the initial starting position (-0.013m), compared to the healthy controls who returned to the initial starting position (0.002m). Nevertheless, as the maximum difference was smaller as the MDC, this is of little clinical value.

Persons with severe KOA had significantly more *pelvic obliquity* at the contralateral side in the first ~20% of the unipodal stance task (Fig 2). The healthy controls showed a small pelvic drop (-1.4˚) on the contralateral side within the first 15% of the waveform, which was subsequently converted into an oblique pelvic angle (1.4˚) at the contralateral side. The persons with severe KOA started already with pelvic obliquity (1.1˚) at the contralateral side and maintained this oblique angle throughout the performance unipodal stance task (Fig 2). However, as the maximum difference was only 2.5 degrees, which is smaller than the MDC, this difference is of little clinical value.

Lastly, the inertial sensor system showed that the persons with severe KOA slightly *flexed their hip* (~3.5˚ flexion) throughout the performance of the unipodal stance task, whereas the healthy controls slightly extended the hip (~3˚ extension) and this difference was greater as the MDC. However, contradicting results were measured by the camera-based system, which showed that both the healthy controls as persons with severe KOA flexed their hip throughout the unipodal stance task (Fig 2). Differences in offset between models were observed in previous studies [27–29]. However, it seems that in this case the hip flexion angles are affected by the orientation of the pelvic sensor (as this is used to determine the hip angles), as the camera-based system measures an average anterior pelvis tilt of 6.5 degrees, while the IMU system measures a posterior pelvis tilt of -1.6 degrees. Therefore, care should be taken to the position of the pelvic sensor, as it is essential for reliable and valid hip and trunk angles.

The camera-based system showed that persons with severe KOA had a significant greater knee adduction angle, from 20 to 22% and in the second half (46–100%) of the unipodal stance

task (Fig 3). Since the knee adduction angle explains, next to lateral trunk lean and pelvic obliquity towards the contralateral side, part of the variation in the knee adduction moment and the fact that knee ab/adduction malalignment is related to the progression of KOA, it is regrettable that these differences were not found by the inertial sensor system [30, 31]. Furthermore, the knee ab/adduction angle is described as a measure of knee instability during the unipodal stance task [9, 10], which is related to reduced physical functioning both in persons with severe KOA and in persons after TKR [4]. Other clinically relevant adaptive movements were also only identified by the camera-based system. persons with severe KOA had more trunk rotation towards the standing leg in the last 8% of the unipodal stance, whereas the healthy controls' trunk did not rotate. In addition, greater knee flexion throughout the waveform was observed for persons with severe KOA compared to the healthy controls (Fig 3). One possible reason for persons with severe KOA to have more knee flexion throughout the unipodal stance task might be related to joint deformation due to KOA, or that it was too painful to further extend the knee [32]. Besides that, other underlying mechanisms, such as loss of proprioception, swelling or the degree of disability could contribute, as it has been shown that reduced joint ROM is not an unidimensional physical characteristic of persons with KOA [32, 33].

It is clear that the inertial sensor system in its current form is not ready for use in the assessment of persons with severe KOA during the unipodal stance task, since several differences between healthy controls and persons with severe KOA were detected by the camera-based system, but not by the inertial sensor system. Specifically regarding the knee adduction and knee flexion angle, this is regrettable since these angles are related to knee joint instability and the progression of KOA. Given that previous results in persons with KOA showed that AP and ML knee accelerations were significantly different from healthy controls and that the AP acceleration was significantly correlated with the COP sway [10], analysis of these signals from inertial sensors could have potential value in the assessment of persons with severe KOA in future.

When interpreting the results of this study some limitations need to be considered. For this study both healthy controls and persons with severe KOA were included to evaluation of the discriminant validity of the joint kinematics measured by the inertial sensor system. It could be argued that the persons with KOA might have difficulties with the performance of the unipodal stance task, although all the included participants performed five repetitions of the unipodal stance task according to the instructions. Additionally, based on the KOOS it was shown in Table 1 that the persons with severe KOA had a significantly lower scores compared to the HC. However, the larger standard deviation for the persons with severe KOA indicates that some of these participants had more associated problems to their knees than others. Inherent to this type of research is a possible selection bias. It is possible that the participants that were recruited were persons which do not cover the full KOA spectrum, given the active assessment protocol. As the patients decided to participant on voluntary basis, it is possible that the patients that are unable to execute the task, or with severe pain didn't want to participate. This was also reflected through the scores of the numeric pain rating scale (0 = no pain; 10 = worst possible pain). Although these results were not included within the manuscript, one participant scored a 9 just after the performance of the 5 repetitions, two participants a score of 5 and the other 16 participants had a score between 0 and 2. Nevertheless, these findings show that for the present study it is feasible to perform an unipodal stance task in persons with severe KOA.

Furthermore, within the present study much effort was done to reduce the measurement error of the inertial sensor system. The inertial sensors were positioned in a standardized manner and strapped in order to avoid soft-tissue artefacts [26]. Additionally, the participants were passively positioned in the correct position, in order to improve the sensor-to-segment alignment during the calibration [34]. On the contrary, the N-pose calibration, which was used to

align the sensor to the segment, presumes that both legs are in full extension. However, as for the present study persons with severe KOA were included, which have limitations in ROM and malalignment of the knee joint, this might affect the calibration and subsequently the calculation of joint angles. Furthermore, it is know that the inertial sensor measurements suffer from integration drift, through (double) integration of the accelerometer and gyroscope signal. The magnetometer is used as a reference, to compensate for this drift [35]. However, as the magnetometer is easily disturbed by ferromagnetic materials, this will affect the sensor-to-segment calibration and the calculation of the orientation and position of the sensor [36]. It is therefore recommended to avoid the proximity of ferromagnetic materials. Nevertheless, completely avoiding the ferromagnetic interference, when measuring inside a laboratory is not possible. New methods have been developed that overcome these constraints by tracking motion without using the magnetometer [37, 38]. However, these methods need further development before they can be implemented in clinical research.

## Conclusion

Assessment of the unipodal stance by means of an inertial sensor system showed that persons with severe KOA have more trunk lean towards the contralateral leg, and more hip flexion compared to healthy controls. Additional discriminating parameters, which are related to postural sway, knee joint instability or the progression of KOA such as a greater knee adduction angle and greater knee flexion angle were measured by the camera-based system. Unfortunately, these differences did not discriminate when measured by inertial sensors, because the detected difference in ROM between healthy control and persons with KOA was smaller as the minimum detectable change or because the construct validity was not sufficient. Further research should investigate the opportunities of knee accelerations, to discriminate between healthy and KOA. Additionally, the reproducibility and accuracy of the inertial sensor measurements should be improved, in order to measure differences in clinically relevant tasks with small movement deviations.

## Supporting information

**S1 Fig.** Waveform comparison between MVN BIOMECH model (red) and musculoskeletal model (blue), from pooled data healthy controls and persons with severe KOA.
(TIF)

**S1 Table. CMC and RMSE of 3D joint kinematics and COM displacement.**
(DOCX)

**S2 Table. Reliability and agreement of 3D joint kinematics and COM displacement.**
(DOCX)

## Acknowledgments

The authors would like to thank Jill Emmerzaal for assistance with the data processing.

## Author Contributions

**Conceptualization:** R. van der Straaten, I. Jonkers, B. Vanwanseele, A. K. B. D. Bruijnes, J. Malcorps, J. Bellemans, J. Truijen, L. De Baets, A. Timmermans.

**Data curation:** R. van der Straaten, M. Wesseling.

**Formal analysis:** R. van der Straaten, M. Wesseling.

**Investigation:** R. van der Straaten, M. Wesseling, A. K. B. D. Bruijnes.

**Methodology:** R. van der Straaten, I. Jonkers, B. Vanwanseele, L. De Baets, A. Timmermans.

**Project administration:** R. van der Straaten.

**Resources:** R. van der Straaten, A. K. B. D. Bruijnes, J. Malcorps, J. Bellemans, J. Truijen.

**Supervision:** I. Jonkers, L. De Baets, A. Timmermans.

**Visualization:** R. van der Straaten.

**Writing – original draft:** R. van der Straaten.

**Writing – review & editing:** R. van der Straaten, M. Wesseling, I. Jonkers, B. Vanwanseele, A. K. B. D. Bruijnes, J. Malcorps, J. Bellemans, J. Truijen, L. De Baets, A. Timmermans.

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
