## [Decision Letter · Decision Letter 0]

6 Sep 2019

PONE-D-19-17344

Discriminant validity of 3D joint kinematics and centre of mass displacement measured by inertial sensor technology during the unipodal stance task

PLOS ONE

Dear Mr van der Straaten,

Thank you for submitting your manuscript to PLOS ONE. After careful consideration, we feel that it has merit but does not fully meet PLOS ONE’s publication criteria as it currently stands. Therefore, we invite you to submit a revised version of the manuscript that addresses the points raised during the review process.

We would appreciate receiving your revised manuscript by Oct 21 2019 11:59PM. To enhance the reproducibility of your results, we recommend that if applicable you deposit your laboratory protocols in protocols.io, where a protocol can be assigned its own identifier (DOI) such that it can be cited independently in the future. For instructions see: http://journals.plos.org/plosone/s/submission-guidelines#loc-laboratory-protocols

We look forward to receiving your revised manuscript.

Kind regards,

Jean L. McCrory, PhD

Academic Editor

PLOS ONE

Journal Requirements:

3. We note that Table 1 includes an image of a [patient / participant / in the study]. 

Additional Editor Comments (if provided):

Reviewers' comments:

Reviewer's Responses to Questions

**Comments to the Author**

1. Is the manuscript technically sound, and do the data support the conclusions?

Reviewer #1: Yes

Reviewer #2: Yes

2. Has the statistical analysis been performed appropriately and rigorously? 

Reviewer #1: I Don't Know

Reviewer #2: Yes

3. Have the authors made all data underlying the findings in their manuscript fully available?

Reviewer #1: Yes

Reviewer #2: Yes

4. Is the manuscript presented in an intelligible fashion and written in standard English?

Reviewer #1: Yes

Reviewer #2: Yes

5. Review Comments to the Author

Reviewer #1: Thank you for the opportunity of reviewing this interesting research aimed at investigating the discriminant validity of three-dimensional joint kinematics and center of mass displacement. The research covers an important topic and I commend the authors for the work performed. However, I have some concerns:

*Statistical methods of validation seem solid, although it would be advisable that they were reviewed by an expert in the field.

Major concerns:

* My major concern is about the methodology/implementation of the proposed test (unipodal stance task). The test had to be performed during at least three seconds, then the participant had to come back to bear the weight in both limbs, and then repeat this sequence for five times. I am familiar with a very similar test, the single/one leg balance test, in which the individual has to stand on one leg and keep balance while bearing the weight in one leg, with the arms placed in the hip, etc. I have used this test in my own research, both with people with KOA in the waiting list for TKR surgery and with healthy matched controls. Healthy controls can probably perform this 5-time sequence test. But I am not sure that every patient with severe KOA actually can. In my experience, a considerable number of individuals cannot even bear the weight for 1-2 seconds. Even less to repeat this sequence for five times without feeling tremendous pain or losing balance. In this line, you mentioned that 19 patients were recruited. Were the 19 patients recruited able to perform the test procedure? Or do you originally recruited more than 19 individuals and excluded some of them?

* The single-leg test may be a suitable test to be used with KOA patients, since the fact that the individual is not able to do it for at least 5 seconds, implies limited balance / stability or high risk of falling. However, the purpose for which this test was used in this research is otherwise: establishing knee instability by means of assessing 3D joint kinematics and the COM displacement. So the fact that patients were or not (and are or not) able to complete the sequence is an important issue

* With respect to the test methodology, maybe I have missed something, since the manuscript has lot of technical information and it is easy to get lost, but... were they allowed to rest between repetitions? Were there a min/max time established? This certainly could influence test performance.

Some other comments:

Abstract.

- Please spell out abbreviations at first mention.

- The authors state one objective in the background section. Then, there are two other objectives in the objective section. Please, clarity and consistency is recommended.

L59. Please, do not use the word elderly, but older adults. This word may be considered derogatory.

L65. The authors introduce limitations of persons with KOA, that is fine. Then the authors state which are the limitations in persons after TKA. However, individuals that have undergone TKA are not being assessed in this work, but individuals with KOA.

L114. The authors mention for the first time that the criteria included “diagnosis of unilateral KOA and awaiting for a total knee replacement (TKR) surgery”. Until this point of the manuscript, the authors had referred to the target population as individuals with KOA. However, if the participants were recruited from a waiting list for TKR surgery, I wonder whether it was because there was a diagnosis indicating a severe/last stage of the condition. If so, I believe the authors should clarify that the target population is not all possible patients with KOA, but patients with severe KOA alone. To do this, information concerning the clinical / radiological diagnosis of eligible participants should also be included, and amend throughout the manuscript to refer to this population.

L120. “Both groups completed the Knee injury and Osteoarthritis Outcome Score (KOOS) to evaluate the extent of symptomatic problems at the knee related to KOA”. This information is in the same paragraph than the inclusion criteria. But, was there a threshold or score in this test used to include/exclude participants? Otherwise this information should be placed in data collection.

* At this point, I have not seen a priori simple calculation, or a post-hoc sensibility analysis to determine whether the sample size was suitable for the study purpose.

Conclusion.

The authors stated three different objectives. Conclusion mention 3D kinematics but no mention to center of mass displacement has been made. Authors also stated in the objectives that “the reliability, agreement and construct validity are assessed to determine the reproducibility and accuracy of the discriminating parameters”. But no conclusion to these parameters was made. Please align conclusions with the objectives. And clarify the objectives.

Reviewer #2: First use of KOA, PwKOA should be written out in abstract. In line 46 "greater as" should be "greater than." In line 51 "before it" should be "before they." On line 141 MVN Biomech Awinda is produced by Xsens, not Vicon Technologies. KAM is only used on line 343 and should be written out.

What method was used to transform the trunk and pelvic angles to account for segment coordinate frame differences?

How was waveform normality verified?

6. PLOS authors have the option to publish the peer review history of their article (what does this mean?). If published, this will include your full peer review and any attached files.

Reviewer #1: Yes: J-M Blasco

Reviewer #2: No

---

## [Author Response · Author response to Decision Letter 0]

22 Oct 2019

A response to the comments of the reviewers is provided in the attached file: "response to reviewers - final".

---

## [Decision Letter · Decision Letter 1]

6 Dec 2019

PONE-D-19-17344R1

Discriminant validity of 3D joint kinematics and centre of mass displacement measured by inertial sensor technology during the unipodal stance task

PLOS ONE

Dear Mr van der Straaten,

Thank you for submitting your manuscript to PLOS ONE. After careful consideration, we feel that it has merit but does not fully meet PLOS ONE’s publication criteria as it currently stands. Therefore, we invite you to submit a revised version of the manuscript that addresses the points raised during the review process.

One of the Reviewers reported that although you address their concerns in the "Responses to Reviewers" letter, the manuscript is essentially unchanged in that the changes did not seem to appear in the manuscript, only in the responses to reviewers.  Please resubmit the modified manuscript with the changes highlighted so that the reviewer can easily see what you changed and that their concerns were addressed.  

We would appreciate receiving your revised manuscript by Jan 20 2020 11:59PM. To enhance the reproducibility of your results, we recommend that if applicable you deposit your laboratory protocols in protocols.io, where a protocol can be assigned its own identifier (DOI) such that it can be cited independently in the future. For instructions see: http://journals.plos.org/plosone/s/submission-guidelines#loc-laboratory-protocols

We look forward to receiving your revised manuscript.

Kind regards,

Jean L. McCrory, PhD

Academic Editor

PLOS ONE

Reviewers' comments:

Reviewer's Responses to Questions

**Comments to the Author**

1. If the authors have adequately addressed your comments raised in a previous round of review and you feel that this manuscript is now acceptable for publication, you may indicate that here to bypass the “Comments to the Author” section, enter your conflict of interest statement in the “Confidential to Editor” section, and submit your "Accept" recommendation.

Reviewer #1: (No Response)

2. Is the manuscript technically sound, and do the data support the conclusions?

Reviewer #1: Yes

3. Has the statistical analysis been performed appropriately and rigorously? 

Reviewer #1: I Don't Know

4. Have the authors made all data underlying the findings in their manuscript fully available?

Reviewer #1: Yes

5. Is the manuscript presented in an intelligible fashion and written in standard English?

Reviewer #1: Yes

6. Review Comments to the Author

Reviewer #1: The article is nice and the methods seem sound. I have the impression that the authors made an effort to answer the reviewer concerns in the resposne letter. However, the original manuscript remains almost unchanged and except for a few small paragraphs, the concerns have not been clarified in the manuscript, but only in the response letter. I would commend to the authors to address the reviewer concerns in the manuscript before acceptance.

7. PLOS authors have the option to publish the peer review history of their article (what does this mean?). If published, this will include your full peer review and any attached files.

Reviewer #1: Yes: JM-Blasco

---

## [Author Response · Author response to Decision Letter 1]

26 Feb 2020

Dear Editor, the response to the reviewers is added in the "response to reviewers - highlights rev1_Final.docx".

---

## [Decision Letter · Decision Letter 2]

17 Apr 2020

Discriminant validity of 3D joint kinematics and centre of mass displacement measured by inertial sensor technology during the unipodal stance task

PONE-D-19-17344R2

Dear Dr. van der Straaten,

We are pleased to inform you that your manuscript has been judged scientifically suitable for publication and will be formally accepted for publication once it complies with all outstanding technical requirements.

With kind regards,

Jean L. McCrory, PhD

Academic Editor

PLOS ONE

Additional Editor Comments (optional):

Reviewers' comments:

Reviewer's Responses to Questions

**Comments to the Author**

1. If the authors have adequately addressed your comments raised in a previous round of review and you feel that this manuscript is now acceptable for publication, you may indicate that here to bypass the “Comments to the Author” section, enter your conflict of interest statement in the “Confidential to Editor” section, and submit your "Accept" recommendation.

Reviewer #1: All comments have been addressed

2. Is the manuscript technically sound, and do the data support the conclusions?

Reviewer #1: Yes

3. Has the statistical analysis been performed appropriately and rigorously? 

Reviewer #1: I Don't Know

4. Have the authors made all data underlying the findings in their manuscript fully available?

Reviewer #1: Yes

5. Is the manuscript presented in an intelligible fashion and written in standard English?

Reviewer #1: Yes

6. Review Comments to the Author

Reviewer #1: (No Response)

7. PLOS authors have the option to publish the peer review history of their article (what does this mean?). If published, this will include your full peer review and any attached files.

Reviewer #1: No

---

## [Editor Report · Acceptance letter]

27 Apr 2020

PONE-D-19-17344R2 

Discriminant validity of 3D joint kinematics and centre of mass displacement measured by inertial sensor technology during the unipodal stance task 

Dear Dr. van der Straaten:

I am pleased to inform you that your manuscript has been deemed suitable for publication in PLOS ONE. Congratulations! Your manuscript is now with our production department. 

With kind regards,

on behalf of

Dr. Jean L. McCrory 

Academic Editor

PLOS ONE